# High-Depth Transcriptome Reveals Differences in Natural Haploid *Ginkgo biloba* L. Due to the Effect of Reduced Gene Dosage

**DOI:** 10.3390/ijms23168958

**Published:** 2022-08-11

**Authors:** Yaping Hu, Petr Šmarda, Ganping Liu, Beibei Wang, Xiaoge Gao, Qirong Guo

**Affiliations:** 1Co-Innovation Center for Sustainable Forestry in Southern China, Nanjing Forestry University, Nanjing 210037, China; 2Department of Botany and Zoology, Masaryk University, Koltlářská 2, 61137 Brno, Czech Republic

**Keywords:** ginkgo, haploid, transcriptome, gene dosage, mechanism

## Abstract

As a representative of gymnosperms, the discovery of natural haploids of *Ginkgo biloba* L. has opened a new door for its research. Haploid germplasm has always been a research material of interest to researchers because of its special characteristics. However, we do not yet know the special features and mechanisms of haploid ginkgo following this significant discovery. In this study, we conducted a homogenous garden experiment on haploid and diploid ginkgo to explore the differences in growth, physiology and biochemistry between the two. Additionally, a high-depth transcriptome database of both was established to reveal their transcriptional differences. The results showed that haploid ginkgo exhibited weaker growth potential, lower photosynthesis and flavonoid accumulation capacity. Although the up-regulated expression of DEGs in haploid ginkgo reached 46.7% of the total DEGs in the whole transcriptome data, the gene sets of photosynthesis metabolic, glycolysis/gluconeogenesis and flavonoid biosynthesis pathways, which were significantly related to these differences, were found to show a significant down-regulated expression trend by gene set enrichment analysis (GSEA). We further found that the major metabolic pathways in the haploid ginkgo transcriptional database were down-regulated in expression compared to the diploid. This study reveals for the first time the phenotypic, growth and physiological differences in haploid ginkgos, and demonstrates their transcriptional patterns based on high-depth transcriptomic data, laying the foundation for subsequent in-depth studies of haploid ginkgos.

## 1. Introduction

In general, high-ploidy plants of the same species tend to exhibit enhanced growth vigor compared to low ploidy ones [1]. This enhanced vigor is reflected by a larger body size [2], faster growth rate [3], and more secondary metabolites [4]. In crops, polyploid technology has been used to develop more advantageous varieties of *Cucumis melo* L. [5], tetraploid hybrid rice [6], decaploid strawberries [7], and so on. Researchers have mainly focused on polyploids and neglected the special value of haploids, as haploid plants are generally dwarfed and have low growth. However, haploids have incomparable advantages in shortening the breeding cycle and accelerating the breeding process [8,9].

Numerous diploid plant species contain spontaneous haploids. However, such haploids appear to be uncommon in gymnosperms [10,11]. Since the exploration of natural haploid plants in *Datura stramonium* in 1922 [12], haploids are becoming a prominent study focus in plant germplasm research. Haploid rarely exists in nature as a species. Almost all the extant haploids are created by the techniques such as anther culture, parthenogenesis, etc. [13]. Pure line haploid germplasm resources are of great significance for breeding improvement and accelerating the process of developing improved varieties. The discovery of haploids has greatly helped in deciphering highly heterozygous plant genomes [14,15]. Therefore, an increasing number of researchers are working on the induction of haploids [16,17,18]. The regeneration system of ginkgo has not yet been established, and various methods of inducing haploids in vitro are also effective in ginkgo [19].

*Ginkgo biloba* L. originated in the Carboniferous period, then underwent quaternary glaciation, and finally survived only in China. From the 18th century, ginkgo has spread from China to Europe via Japan and Korea [20,21]. Ginkgo is widely used in daily production and life as an important economic tree species. For instance, flavonoids play an important role in *G. biloba* extract (GBE), which is widely employed in clinical treatment and is one of the main sources of commercial value for ginkgo [22,23]. Ginkgo flavonoid content is affected by many factors, such as fertilization [24], salt [25], temperature [26], light [27], etc. Many regulatory genes and transcription factors (TFs) have been identified to resolve the regulatory network of ginkgo flavonoids [28,29,30]. However, all these studies were based on diploid ginkgo, as no one had found ginkgo of another ploidy until 2016. We first discovered natural ginkgo haploids using flow cytometry and investigated their cytological properties in 2016 [31,32]. The discovery of ginkgo haploids shattered the earlier perception that ginkgo is only diploid. Exploring the differences in haploid compared to diploid ginkgo is important for understanding the effects of gene dosage on gene expressions.

In this study, we obtained high-depth transcriptomic data of natural haploid and diploid ginkgos and used these data to reveal the underlying mechanisms of many differences in haploid ginkgos. This was the first report of transcriptomic differences following the discovery of natural haploid ginkgo material.

## 2. Results

### 2.1. Haploid Ginkgo Exhibits Weak Growth, Biomass Accumulation and Secondary Metabolic Synthesis

Just as most polyploid plants tend to exhibit relatively high growth potential, haploid ginkgos are weaker than diploids in growth, photosynthesis, and secondary metabolite accumulation (Figure 1). The haploid ginkgo leaf area (7.9 ± 0.5 cm^2^) was significantly smaller than that of the diploid (17.6 ± 1.1 cm^2^) when the functional leaves matured in August (Figure 1). Similarly, the net photosynthetic rate and flavonoid concentration of haploid ginkgo were substantially lower than that of diploids (*p* < 0.05). The net photosynthetic rate of haploid ginkgo was 36.1% that of diploid. The lack of photosynthetic capacity leads to weaker biomass accumulation in haploid ginkgo morphology. We used flow cytometry (FLC) to identify the ploidy of the haploid and diploid ginkgo cultivars (Appendix A).

### 2.2. Transcriptomic Analysis Overview

To further explain the differences in transcriptional regulation among different ploidy ginkgos, we selected haploid ‘Rocky’ and diploid ‘Maronia’ with large differences in flavonoid content for transcriptome analysis. Six RNA libraries were generated and analyzed (H1, H2, and H3 for haploid ginkgo, and D1, D2, and D3 for diploid ginkgo). RNA-Seq generated 149.82 GB of clean data (Q30 > 94%) with an average of 24.97 Gb per sample (Table 1). In order to explore the transcriptional differences between haploid and diploid ginkgo to the greatest extent, the amount of transcriptome data in this study was much larger than that of conventional transcriptomes. There were 486.71 million high-quality clean readings collected in total, ranging from 73.31 to 89.43 million per sample. The mapped clean reads were assembled into 29,036 genes (Appendix A) after alignment with the latest ginkgo reference genome. Among these genes, a total of 2452 genes (Appendix A) were identified as differentially expressed (DEGs), with 1149 upregulated and 1303 downregulated (haploid vs. diploid).

### 2.3. Gene Ontology (GO) and Kyoto Encyclopedia of Genes and Genomes (KEGG) Analysis of DEGs

Overall, 34 biological process (BP) phrases, 9 cellular component (CC) terms, and 35 molecular function (MF) terms were found to be enriched in 2485 DEGs. Among the top 20 GO-terms of enrichment, ‘detoxification’, ‘cell wall organization’, and ‘external encapsulating structure organization’ were the most enriched in the BP categories (Figure 2A). ‘Peroxidase activity’, ‘oxidoreductase activity, acting on peroxide as acceptor’, and ‘ADP binding’ were the top three subcategories of enrichment degree in the MF category. In the CC category, the subcategory with the highest enrichment degree was ‘cell wall’, followed by ‘external encapsulating structure’ and ‘extracellular region’.

KEGG analysis of transcriptomics showed that 502 candidate genes with pathway annotations were enriched in 106 KEGG pathways (Appendix A). ‘Monoterpenoid biosynthesis’ was the most substantially enriched pathway, with five up-regulated and three down-regulated genes (Figure 2B), followed by ‘Flavone and flavonol biosynthesis’ and ‘Cutin, suberirne and wax biosynthesis’.

### 2.4. Differential Analysis of Photosynthesis Metabolic Pathway

The net photosynthesis plays a crucial role in plant growth and development. Compared to diploids, haploid ginkgo has seven DEGs enriched in the photosynthesis pathway, six of which were down-regulated and one up-regulated in expression (Figure 3). F-type ATPase, widely found in chloroplast vesicle-like membranes, is known as the world’s smallest and most sophisticated molecular motor [33]. Two key genes regulating F-type ATPase activity, *atpF* and *atpA*, were significantly down-regulated in haploid ginkgo. Additionally, in the reaction process of photosystem II, *rpoA* and *PSBW* were also significantly down-regulated. However, not all genes in photosystem II were down-regulated in haploid ginkgo, such as *petH*. Based on these analyses, we tentatively hypothesized that the weaker photosynthetic capacity of haploid ginkgo was possibly due to the down-regulation of some key genes in the photosynthetic response process.

Enrichment analysis based on traditional hypergeometric tests often requires the use of significantly different gene set data. When changes in the expression of individual genes are weak, little or no results may be obtained based on traditional enrichment analysis. Gene set enrichment analysis (GSEA) can effectively make up for the lack of effective information for minor genes in traditional enrichment analysis, and more comprehensively explain the regulatory role of a functional unit [34,35]. We set up all genes in the photosynthesis pathway as a gene set to analyze the overall expression of photosynthesis genes in haploid ginkgo (Figure 4). The defined core set of regulatory genes was shown in Appendix A. We could find that the expression of genes related to plant photosynthesis showed a significant decrease in haploid ginkgo. Therefore, we can clearly state that the lower photosynthesis capacity of haploid ginkgo was due to the low dosage effect of photosynthesis-related genes.

### 2.5. Differential Analysis of Glycolysis/Gluconeogenesis Pathway

The primary physiological function of sugars is to provide the body with the energy it needs for its vital activities. Gluconeogenesis is the main way in which organisms obtain energy and is a common stage through which all organisms must undergo gluconeogenesis. A total of 17 DEGs were enriched in the glycolysis/gluconeogenesis pathway of haploid ginkgo, of which 11 were up-regulated and 6 were down-regulated in expression (Figure 5). In terms of the number of DEGs in the glycolysis/gluconeogenesis metabolic pathway alone, the number of up-regulated DEGs expressed in haploid ginkgo was even greater than the number of down-regulated expression. However, it was found by GSEA that the entire set of genes regulating glycolysis/gluconeogenesis metabolism containing 140 genes showed a pattern of reduced expression in haploid ginkgo (Figure 6). The entire gene set can be found in Appendix A.

### 2.6. Differential Analysis of Flavonoid Biosynthesis Pathway

Just as haploid ginkgos exhibit lower photosynthesis capacity and growth potential, haploids also had lower flavonoid content than diploids. Unlike the model plants where the flavonoid biosynthesis pathways were relatively well understood, the regulatory network of flavonoid biosynthesis in ginkgo is not yet fully defined. Some key regulatory genes in the flavonoid biosynthesis pathway of haploid ginkgo were down-regulated in expression, such as *F3H*, *FLS*, *HST*, and *CYP75A* (Figure 7). However, the key genes regulating the production of 5-Deoxyleucocyanidin, Leucopelargonidin, and Luteoforol, *DFR*, was up-regulated in haploid ginkgo. In addition, the key gene, *LAR*, which regulates Afzelechin and (+)-Gallocatechin production, was also down-regulated in expression.

Although the expression of DEGs in the haploid ginkgo flavonoid biosynthesis pathway was inconsistent, we could see by GSEA that the genes related to flavonoid biosynthesis showed an overall trend of down-regulated expression (Figure 8). The defined core set of regulatory genes was shown in Appendix A. However, it was still possible to find that there are some genes that do not follow this trend. This also indicates that there are still unexplored corners of the ginkgo flavonoid regulatory network.

### 2.7. Enrichment Analysis of Up-Regulated DEGs in Haploid Ginkgo

The up-regulated genes in haploid ginkgo accounted for 46.7% of the entire differentially expressed gene set. We performed GO enrichment analysis on the up-regulated genes in haploids and found that the most significant enriched term for these genes was ‘diaminobutyrate acetyltransferase activity’ (Figure 9A). The most significantly enriched KEGG pathway of up-regulated DEGs was ‘Plant-pathogen interaction’, followed by ‘Protein processing in endoplasmic reticulum’ and ‘Glutathione metabolism’ (Figure 9B). GSEA results on up-regulated expression of DEGs in haploid ginkgo showed that only two metabolic pathways were significantly enriched, ‘Protein processing in endoplasmic reticulum’ and ‘MAPK signaling pathway—plant’ (Figure 10).

### 2.8. An Overall Trend of Down-Regulated Expression of DEGs in Haploid Ginkgo

We further performed GSEA on all the major enriched pathways of DEGs in the whole transcriptome database, and the results all remained consistent. A total of 21 KEGG metabolic pathways were significantly enriched and showed a trend of down-regulated expression (Figure 11, Appendix A). The top three significantly enriched pathways were ‘Photosynthesis’, ‘Fatty acid biosynthesis’, and ‘Cutin, suberine and wax biosynthesis’.

### 2.9. Transcript Level and Quantitative Real-Time PCR (qRT-PCR) Validation of mRNA in Haploid and Diploid Ginkgo

Fourteen randomly selected DEGs in the ‘Flavonoid biosynthesis’ and ‘Phenylpropanoid biosynthesis’ pathways were used to perform qRT-PCR to verify that the transcription group data is valid and reliable (Figure 12). These genes’ real-time fluorescence expression trends were consistent with the transcriptome data, showing that our transcriptome data was credible. Significantly, the expression of several genes in the ‘Phenylpropanoid biosynthesis’ pathway was significantly higher in haploid ginkgos than in diploids, such as evm.TU.chr4.2086, evm.TU.chr10.1202 and evm.TU.chr10.109. The ‘Phenylpropanoid biosynthesis’ pathway, as an important upstream regulatory pathway of ‘Flavonoid biosynthesis ‘, is one of our priorities and the next step to be taken.

## 3. Discussion

As with most species, polyploid plants are taller and grow faster, and haploid plants are relatively short and weak, as is the case with ginkgo. Polyploid ginkgos have larger stomata and cells [32]. Larger cells and more cellular content can accumulate more energy for their growth and development. Qualitative differences in morphology, growth, and physiology and biochemistry among ginkgos of different ploidy follow the laws of ploidy, as has been demonstrated on Liriodendron *sino-americanum* [36], *Eucommia* [37], rice [38], *Rhododendron fortunei* [39], and *Thymus vulgaris* [40]. However, this is not absolute; for example, the size of pure tetraploid apples (*Malus* × *domestica*) and *Citrus limonia* are smaller compared to their diploid counterparts [41,42].

The increasing amount of gene expression data in ploidy studies has stimulated scientists to speculate on the regulatory mechanisms driving these gene expression changes. Plant polyploidization may cause DNA sequence mutations such as chromosomal rearrangements and gene loss, DNA modifications, and changes in gene expression [43,44]. In the present study, haploid ginkgos showed different degrees of DEGs compared to diploid ginkgos. We focused on metabolic pathways that exhibit apparent differences in photosynthesis, glycolysis/gluconeogenesis, and flavonoid biosynthesis. Although the extent of changes in the expression of DEGs in these pathways was not entirely consistent, the GSEA showed an overall trend of down-regulation in haploid ginkgo. Haploid ginkgos present a marked reduction in gene dosage compared to diploids. Studies have confirmed that DNA methylation and small RNAs play an important role in the expression of different genes [45].

However, within this overall trend of down-regulated expression, up-regulated expression of genes in haploid ginkgo accounted for 46.7% of the total differentially expressed genes. However, these DEGs were only significantly enriched in the ‘Protein processing in endoplasmic reticulum’ and ‘MAPK signaling pathway–plant’ pathways by GSEA. ‘MAPK signaling pathway–plant’ is widely involved in plant immunity and environmental stress response. [46]. Haploids are less stable in the natural environment and less resistant to adversity than diploids. Genes in the ‘MAPK signaling pathway–plant’ pathway were up-regulated in haploid ginkgo, signaling that haploid ginkgo was trying to adapt to the habitat.

The traits of an organism are regulated by genes, and transcription is an important but not all-inclusive part of the transmission of genetic information. Proteins are the executors of genetic information. Combining the proteomics of haploid ginkgo may better explain the reasons for these differences.

We are more interested in haploid ginkgo than in other ploidy types, as haploids are perfect materials for perfecting the highly heterozygous, oversized mega-genome of ginkgo [47]. Our next step is to further improve the quality of the ginkgo reference genome by using haploid material. The use of haplotype genomes to resolve complex trait regulatory mechanisms and fine localization of regulatory loci has achieved significant success, especially in highly complex genomic tree species [14,48,49]. Deciphering the haplotype genome of ginkgo will greatly help to resolve its important metabolic mechanisms, such as flavonoids, lactones, and polyprenols.

In summary, the difference between haploid and diploid ginkgo was mainly related to the reduced dosage of relevant genes. This work not only laid the foundation for the study of haploid ginkgo systems, it was also the first report to fully exploit haploid ginkgo genetic differences.

## 4. Materials and Methods

### 4.1. Plant Material and Sample Preparation

Haploid and diploid ginkgo plants were obtained from the Botanical Garden of Masaryk University. We grafted scions of 3 haploid ginkgo cultivars (‘Clica’, ‘Rocky’, and ‘Baldii’) and 3 diploid ginkgo cultivars (‘David’, ‘Tit’, and ‘Maronia’) on the same ginkgo cultivar in the Ginkgo Germplasm Resource Garden of Nanjing Forestry University (119.19 E, 31.58 N) in March 2020. All the scions were of the same length and had two buds. All scions were harvested from the first-year lateral branches of each maternal plant. The environmental conditions for the growth of the plants in the entire germplasm nursery were maintained in a group without significant adversity.

Ginkgo leaf samples for RNA isolation was collected after grafting the pants. The leaf samples were stored at −80 ℃ after flash freezing in liquid nitrogen.

### 4.2. RNA Sequencing and Data Analysis

Total RNA was extracted using the TRIzol reagent (Invitrogen Scientific, Inc., Carlsbad, CA, USA) in accordance with the manufacturer’s protocol. The quality of the total RNA of the sample was detected using an Agilent 2100 Bioanalyzer (Agilent Technologies, Inc., Santa Clara, CA, USA) with an RNA integrity number (RIN) value of ≥7 required for all samples. After sequencing, the sequences were filtered to obtain high quality clean reads using fastp by removing the low quality bases and adapters in the sequences. Thus, data were further filtered using fastp [50] (version 0.18.0) to acquire high-quality clean reads, and the remaining clean reads were mapped to the most recent ginkgo reference genome [47] using HISAT2.4 [51]. The ribosomal RNA (rRNA) database was mapped using the short read alignment program Bowtie2 [52] (version 2.2.8). The mapped reads of each sample were assembled using StringTie [53,54] v1.3.1, using a reference-based approach. DESeq2 [55] software is used to analyze RNA differential expression between two separate groups (and edgeR [56] between two samples).

The gene sequencing method used for GSEA was Signal2Noise. FDR < 0.25 and normalized *p* < 0.05 were the filter conditions, and other parameter settings were kept as default values.

Bioinformatic analysis was performed using Omicsmart, a real-time interactive online platform for data analysis (http://www.omicsmart.com) (accessed on 2 May 2022).

The raw sequencing data from Illumina were uploaded to the NCBI Short Reads Archive database with the accession number SRP337737.

### 4.3. Determination of Flavonoid Content and Photosynthetic Rate of Ginkgo

We collected fully unfolded mature leaves and determined ginkgo flavonoid content using high-performance liquid chromatography (HPLC) [27]. First, the ground blade sample (0.5 g) was refluxed for 2 h with 50 mL of chloroform in a Soxhlet extractor and then evaporated to dryness. The dried residue was then extracted with 50 mL of methanol for 4 h, at 80 °C. After cooling to room temperature, the eluent was diluted to 50 mL with methanol, and the concentrations of quercetin, kaempferol, and ISorhamNetin were measured by HPLC. The following conditions were set for HPLC (Waters 1525, Canby, OR, USA): The mobile phase was a 1.0 mL/min solution of methanol and 0.4 percent H3PO4 (56:44, *v/v*); the column temperature was 30 °C; the sample was detected at 360 nm. The following conditions were set for HPLC (Waters 1525, USA): The mobile phase was a 1.0 mL/min solution of methanol and 0.4 percent H3PO4 (56:44, *v/v*). Total flavonoid content = (quercetin + kaempferol + isorhamnetin content) × 2.51 [57].

The photosynthetic rate of ginkgo was measured by CIRAS-3 (PP Systems, Amesbury, MA, USA) on a clear and windless morning in August 2021. The functional leaves of the same position were selected for each plant, and the determination was repeated three times. The photosynthetically effective radiation of the LED light source was 1200 umol∙m^−2^∙s^−1^, consisting of 90% red light, 5% blue light and 5% white light.

### 4.4. Quantitative Real-Time PCR (qRT-PCR)

We used qRT-PCR to confirm the expression of 14 DEGs in the flavonoid metabolic and photosynthesis metabolic pathways. Detailed primer information is provided in Appendix A. The 2^−∆∆Ct^ technique was used to calculate relative expression levels [58]. *GbCHS* (F: CAAGCGCATGTGCGACAAGT, R: CACCTCCACCACCACCATGT) was used as an endogenous control [59]. Each reaction in every experiment was repeated three times.

## 5. Conclusions

In this study, we found that haploid ginkgos exhibited lower growth potential, photosynthesis and flavonoid accumulation capacity through homogeneous garden trials of haploid and diploid ginkgos. The deepened transcriptome database revealed 2485 down-regulated DEGs expressed in haploid ginkgo, representing 53.3% of the whole DEGs and 8.9% of the whole genome. Although key genes in these metabolic pathways were expressed both up- and down-regulated in haploids, the gene dosage showed an overall decreasing trend by GSEA. Reduced gene dosage was the main factor leading to various differences in haploid ginkgos. This study provides a theoretical basis for the in-depth study and resource utilization of haploid ginkgo.

## Figures and Tables

**Figure 1 ijms-23-08958-f001:**
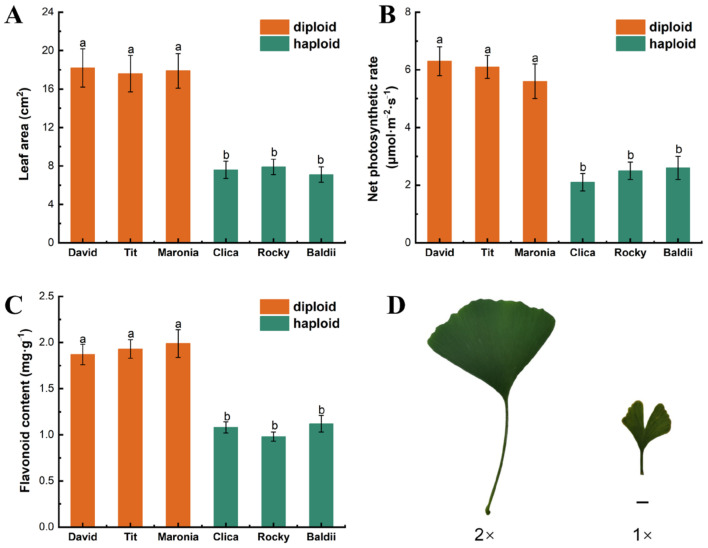
Comparison of morphology, growth, photosynthesis and flavonoid content of haploids and diploids. Schemes follow the same formatting. (**A**) Leaf area. (**B**) The net photosynthetic rate of the leaves. (**C**) Leaf flavonoid content. (**D**) Haploid and diploid leaf morphology. Bars with different letters indicate significant differences at *p* < 0.05 according to Duncan’s test; 1×: haploid plants; 2×: diploid plants. Bar = 1 cm.

**Figure 2 ijms-23-08958-f002:**
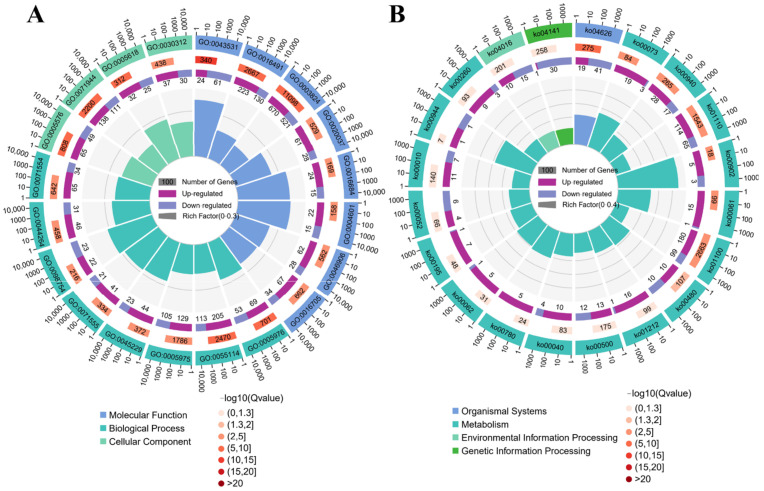
DEG enrichment analysis utilizing GO and KEGG pathways. (**A**) GO classifications of DEGs. (**B**) KEGG classifications of DEGs. The first circle from outside and inside is the pathway of the top 20 enrichment terms, and the number outside the circle is the gene number scale. The second circle represents the number and Q value of the pathway in the background gene. The third circle represents the number and ratio of up- (light purple) and down-regulated (dark purple) genes. The fourth circle represents the Rich Factor value of each pathway. GO:0043531, ADP binding. GO:0005976, polysaccharide metabolic process. GO:0016491, oxidoreductase activity. GO:0003824, catalytic activity. GO:0003824, catalytic activity. GO:0005618, cell wall. GO:0071944, cell periphery. GO:0055114, oxidation-reduction process. GO:0030312, external encapsulating structure. GO:0005975, carbohydrate metabolic process. GO:0016684, oxidoreductase activity, acting on peroxide as acceptor. GO:0004601, peroxidase activity. GO:0004601, peroxidase activity. GO:0045229, external encapsulating structure organization. GO:0020037, heme binding. GO:0044264, cellular polysaccharide metabolic process. GO:0071554, cell wall organization or biogenesis. GO:0098754, detoxification. GO:0006633, fatty acid biosynthetic process. GO:0016798, hydrolase activity, acting on glycosyl bonds. ko04626, Plant-pathogen interaction. ko00073, Cutin, suberine and wax biosynthesis. ko00940, Phenylpropanoid biosynthesis. ko01110, Biosynthesis of secondary metabolites. ko00061, Fatty acid biosynthesis. ko00902, Monoterpenoid biosynthesis. ko01100, Metabolic pathways. ko00480, Glutathione metabolism. ko00480, Glutathione metabolism. ko00500, Starch and sucrose metabolism. ko00040, Pentose and glucuronate interconversions. ko00062, Fatty acid elongation. ko00062, Fatty acid elongation. ko00780, Biotin metabolism. ko00195, Photosynthesis. ko00052, Galactose metabolism. ko00010, Glycolysis/Gluconeogenesis. ko04016, MAPK signaling pathway-plant. ko00944, Flavone and flavonol biosynthesis. ko00130, Ubiquinone and other terpenoid-quinone biosynthesis.

**Figure 3 ijms-23-08958-f003:**
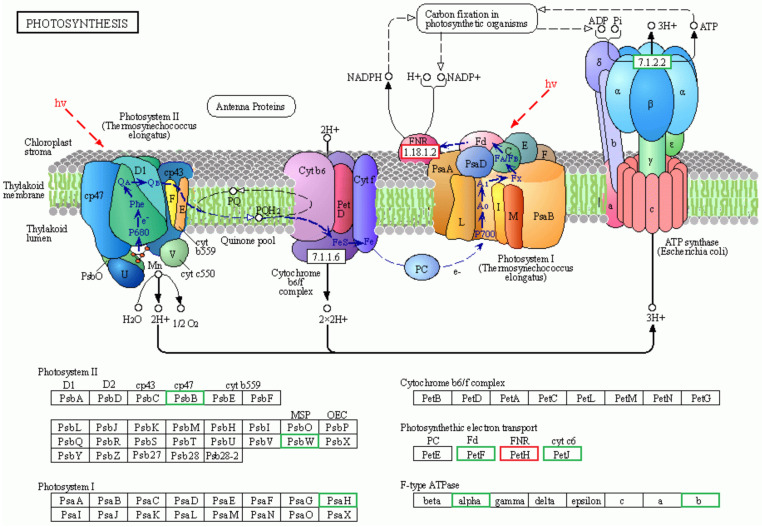
Schematic diagram of DEGs in the photosynthesis pathway of haploid ginkgo. Green indicated down-regulated expression and red indicated up-regulated expression.

**Figure 4 ijms-23-08958-f004:**
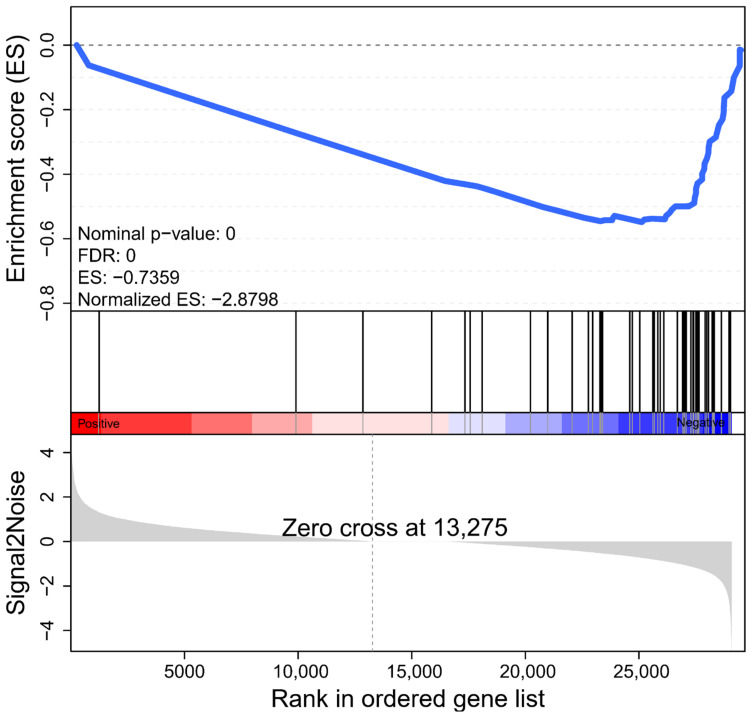
GSEA of all genes in the photosynthesis pathway. The folded line is the enrichment score (ES) of each gene, and the peak value is the ES of the gene set. The horizontal axis represents each gene in this gene set and corresponds to a vertical line similar to a barcode. ES: Enrichment Score. Normalized ES: Normalized ES value after correction. FDR: *p*-value after correction for multiple hypothesis testing. Probability estimates of possible false positive results for Normalized ES. The smaller the FDR, the more significant the enrichment is. Core genes are the genes before the peak for gene sets with positive ES. The core genes are the genes after the peak for the gene set with negative ES. Overall, if genes are enriched at the top of the curve, the gene set is up-regulated expression trend, and conversely, if they are enriched at the bottom, the expression trend is down-regulated.

**Figure 5 ijms-23-08958-f005:**
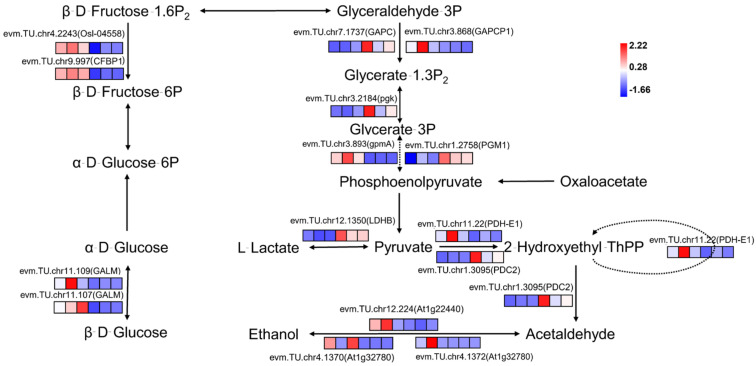
Schematic diagram of DEGs in the glycolysis/gluconeogenesis pathway of haploid ginkgo. The three squares on the left of the gene expression heatmap represent haploid ginkgo, and the three squares on the right represent diploid ginkgo.

**Figure 6 ijms-23-08958-f006:**
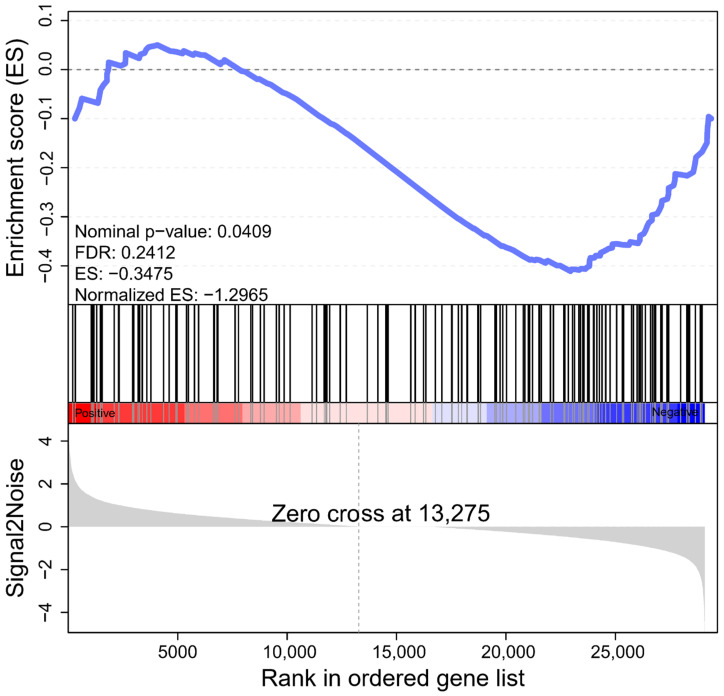
GSEA of all genes in the glycolysis/gluconeogenesis pathway.

**Figure 7 ijms-23-08958-f007:**
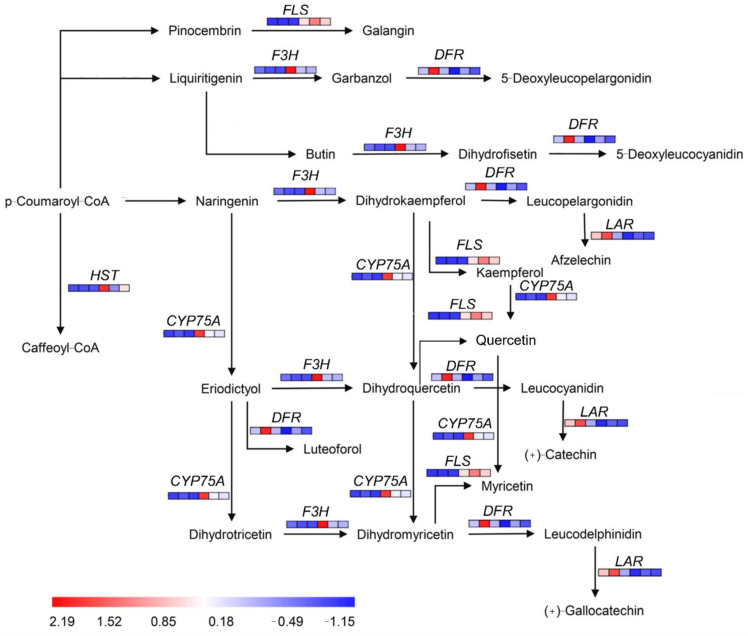
Schematic diagram of DEGs in the flavonoid biosynthesis pathway of haploid ginkgo.

**Figure 8 ijms-23-08958-f008:**
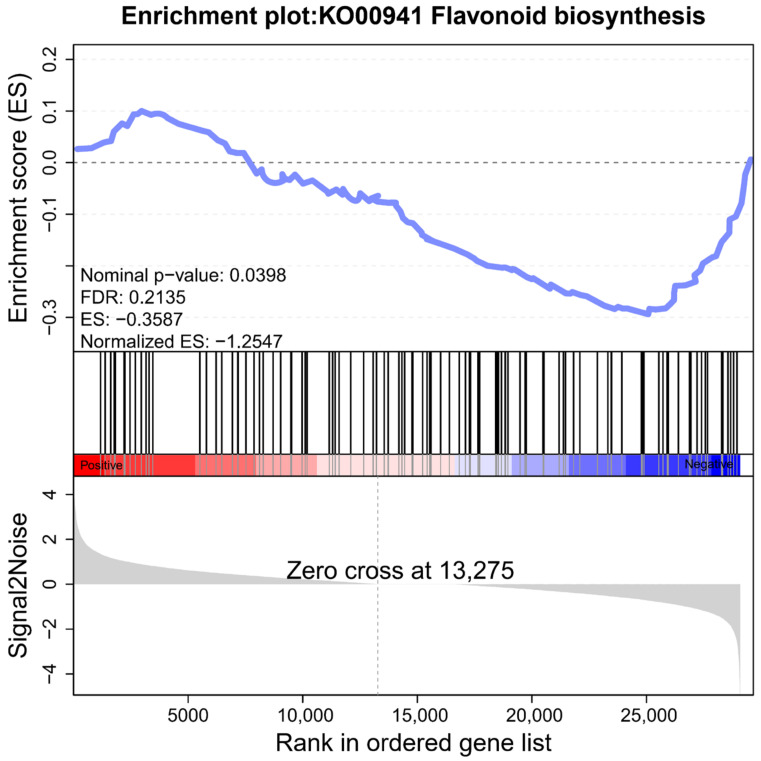
GSEA of all genes in the flavonoid biosynthesis pathway.

**Figure 9 ijms-23-08958-f009:**
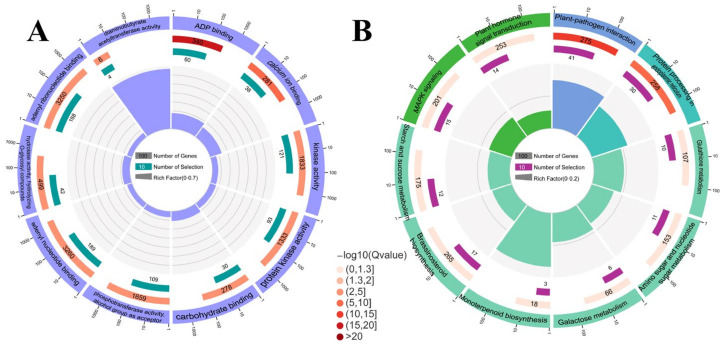
The up-regulated DEGs enrichment analysis utilizing GO (**A**) and KEGG (**B**) pathways.

**Figure 10 ijms-23-08958-f010:**
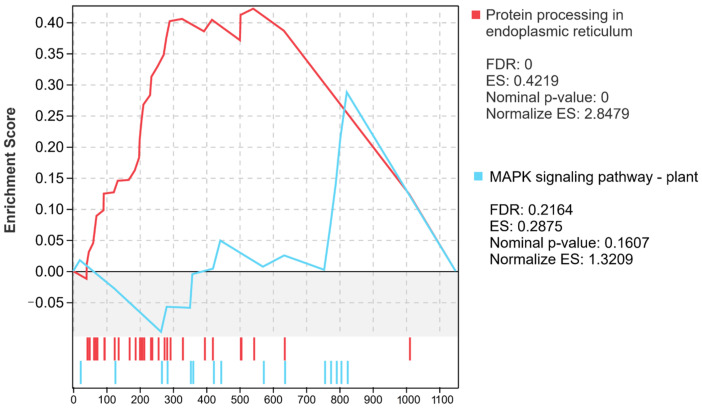
GSEA of up-regulated DEGs in haploid ginkgo.

**Figure 11 ijms-23-08958-f011:**
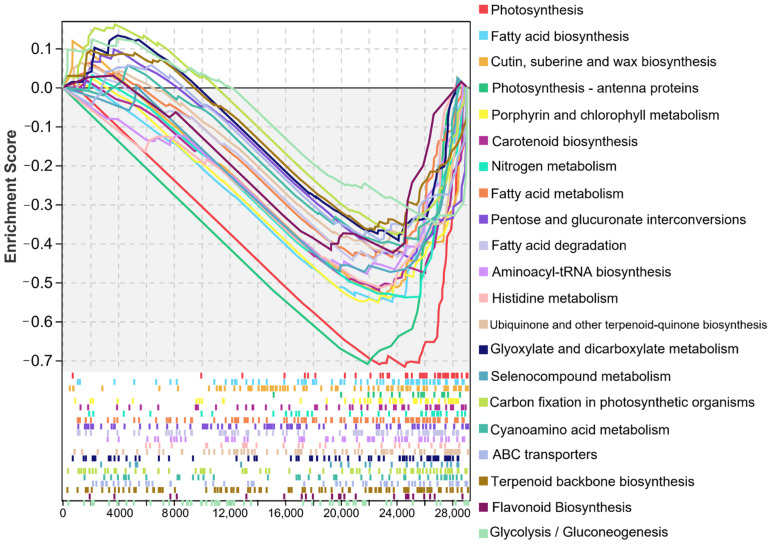
The metabolic pathways significantly enriched in GSEA.

**Figure 12 ijms-23-08958-f012:**
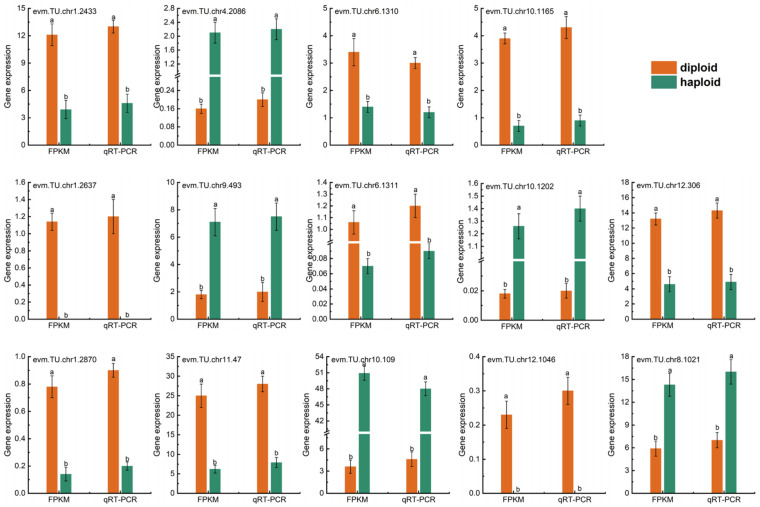
Validation of 14 chosen genes by qRT-PCR in haploid and diploid ginkgo leaves. Bars with different letters indicate significant differences at *p* < 0.05 according to Duncan’s test.

**Table 1 ijms-23-08958-t001:** An overview of the sequencing quality of six ginkgo RNA libraries.

Sample	Clean Reads	Clean Data (bp)	Q20 (%)	Q30 (%)	GC (%)
H1	84,053,070	12,606,818,190	97.84	94.66	44.72
H2	74,996,246	10,743,266,079	97.76	94.37	44.65
H3	73,313,140	10,496,771,635	97.87	94.66	44.55
D1	89,428,662	13,348,479,805	97.80	94.55	44.39
D2	82,482,584	12,311,759,601	97.79	94.46	44.48
D3	82,336,494	12,322,466,518	97.91	94.79	44.47

## Data Availability

The raw data supporting the conclusions of this article will be made available by the authors, without undue reservation.

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
