# Peer review of "High-Depth Transcriptome Reveals Differences in Natural Haploid *Ginkgo biloba* L. Due to the Effect of Reduced Gene Dosage"

_ijms, 2022, doi:10.3390/ijms23168958_

Round 1

Reviewer 1 Report

I think the authors need to redesign the work for I didn't see the advantages of high depth sequencing nor the importance of haploidy.

In fact, your work is mainly about the comparison of transcriptome between the haploids and the diploids of ginkgo. But there is a big problem in your comparison, that is, the haploids and diploids you used are completely different cultivars, so it is impossible to determine whether it is caused by ploidy or genotype. Additionally, the phenotype of photosynthesis is incomplete and lacks description in "Materials and methods" section. So in my opinion, this manuscript has not yet met the criteria for submission.

Reviewer 2 Report

This study uses high-depth transcriptomes to uncover the potential metabolic difference between natural haploid and diploid ginkgo. The initiative of this research is great and will contribute tremendously to the genome and molecular breeding research. However, my major concern of this study is the quality of data presentation and analysis. Many figures are presented without enough explanation in the legends and most enriched pathways are not covered at all in the manuscript (and not even directly available in the figures). The authors' main conclusion is that most enriched pathways are down-regulated in the haploid than the diploid. However, since almost 50% DEGs are actually up-regulated in the haploid, the metabolic difference between haploid and diploid ginkgo should be a lot more complicated than what the authors claimed. This manuscript uses GSEA as a major tool for functional analysis. However, the p-values and q-values provided by GSEA do not support the authors' conclusion that the glycolysis and flavonoid biosynthesis is significantly enriched and down-regulated in haploid than diploid. Therefore, the conclusions presented in this study are overly simplistic and without valid statistical support, preventing this study published in its current shape. 

Following are my detailed comments for this manuscript:

1. line 94-95, the reads were mapped to the latest ginkgo reference genome. However, no reference was provided here (through provided later in the methods) and the associated supplemental file is the gene expression datasets along with the gene annotation, rather than the file of reference genome (e.g. gene models, chromosome coordinates, sequences, etc.). 

2. The authors didn't provide the list for the 2485 DEGs, which would be important results if the readers wanted to repeat the work in this study. 

3. line 101-113 and Fig.2: in the texts the authors described the enriched GO terms and KEGG pathways without referencing the id of GO or KEGG, and fig.2 only provided the id without providing the description. The texts and figure each provides half information, making it difficult for readers to double check. Moreover, it would be better to provide the names of all enriched terms / pathways rather than just providing the ids, as names provide a lot more information than the ids. Though the readers could always check the ids themselves, but this should be the authors' work rather than the readers'.

4. line 110-113: so there were 8 DEGs belonging to "monoterpenoid biosynthesis", what about "flavone and flavonol biosynthesis" and "cutin, suberin, and wax biosynthesis"?

5. line 119: the authors used the word "enriched", is photosynthesis really enriched among DEGs? I cannot tell from fig2 (please check my comment #3). And a similar problem persists in line 150 for glycolysis pathway.

6. Fig.4: The authors need to explain more in the legend how to interpret the figure. For example, what is the meaning of enrichment score and signal2noise? How does the figure support the author's conclusion (that photosynthesis is in general down-regulated in haploid ginkgo)? Not every reader is familiar with GSEA, and the figure should include all necessary information to help the readers understand without guessing the meaning of it. The same comment applies to Fig.6 and Fig.8.

7. Figs.4, 6, and 8: the FDR-corrected q-value for photosynthesis is 0, for glycolysis is 0.27, and for flavonoid biosynthesis is 0.37. The q-values for the latter two pathways are too high to be considered as significant, directly against the authors' conclusion.

8. Fig.5: I assume the six grids in the heatmap representing the six ginkgo libraries? The authors need to clarify this in the legend, including which grids are from the haploid and which are from diploid. And the same applies to Fig.7.

9. Fig.9: Though the expression difference between haploid and diploid is quite substantial, the p-values still should be indicated as valid statistical support.

10. Fig.10: it should be a part of the results, please move it to the results rather than discussion. Moreover, what are those major enriched pathways? The authors only provide the KEGG id without describing the pathways themselves, making this figure contain very little actually useful information.

11. line 224, there are 46.7% DEGs actually up-regulated in haploid, and this is quite a substantial amount. However, this study has covered very little on this aspect. I suppose separate enrichment analyses of pathways for up- and down-regulated genes may reveal more regarding the metabolic difference between haploid and diploid ginkgo. For example, which pathways are up-regulated in haploid than diploid, and which are down-regulated? Actually, there have been publications recommending separate enrichment analyses as they could identify more pathways pertinent to the phenotypic difference.

Reviewer 3 Report

Good work done. This research will open new insights into ginkgos research. I have a few comments which are given below

Line 33 and 34

“In crops, polyploid technology has been used to cultivate more advantageous varieties of Cucumis melo L. [5]”

Comment: Usage of “cultivate” gives different meaning it should be “develop”

Line 38

Comment: breeding years can be changed to breeding cycle

Line 45

“breeding improvement and accelerating the process of improved varieties. The discovery”

Comment: breeding improvement and accelerating the process of developing improved varieties. The discovery

line 66

“This is the first time that genomic differences in haploids have  been explored following the discovery”

Comment: Instead of using time report can be used and the sentence can be modified accordingly

Line 118

The intensity of photosynthesis plays a crucial role in plant growth and development.

Comment: Instead of “intensity of photosynthesis” “net photosynthesis or photosynthetic rate” can be used.

Line 126, 127, 128

Based on these analyses, can we prove that the weak photosynthetic capacity of haploid ginkgo was caused by the down-regulation of some key genes during the photosynthetic reaction?

Comment: Instead framing as a question, the authors can put forth a hypothesis

Line  213 and 214

“We focused on metabolic pathways that explain the observed differences, such as photosynthesis metabolic, glycolysis / gluconeogenesis, and flavonoid biosynthe-“

Comment: Consider modifying as follows, “We focused on metabolic pathways that exhibit apparent differences in photosynthesis, glycolysis / gluconeogenesis, and flavonoid biosynthe-“

Line 220

What drives differential gene expression across ploidy? Studies have confirmed that DNA

Comment: Interrogative sentences can be avoided. It can be modified appropriately.

Line 239

haploid ginkgo system, which will be the first to fully exploit the practical value of haploid

Comment: “first report” can be used instead of “first” and modify the sentence accordingly

Line 250 and 251

“Ginkgo leaves were collected after grafting and stored in liquid nitrogen flash freez-
ing at −80°C for RNA extraction at the end of August 2021”

Comment: It can be like “Ginkgo leaf samples for RNA isolation was collected after grafting the pants. The leaf samples were stored at -80 after flash freezing in liquid nitrogen”

Line 253 and 254

“Raw reads acquired from sequencing equipment may contain adapters or low-qual-

ity bases that impair later assembly and analysis. Thus, data were further filtered using”

Comment: Usage of “may” and “equipment” can be avoided. Instead directly it can be mentioned as “after sequencing the sequences were filtered to obtain high quality clean reads using fastp by removing the low quality bases and adapters in the sequences”.

Fig 1.

Dot can be removed between the units

Questions

Did the author measure photosynthesis per unit leaf area? If so, is there any difference between the photosynthesis per unit leaf area?

Please include the age of the plant and growth conditions where the plants are grown as RNA expression is highly influenced by the environment.

All the best

Thank you

Round 2

Reviewer 1 Report

I didn't see the exact values of these haploids in this article, especially their importances in Ginkgo breeding. On the contrary, these haploids are relatively poor in all traits except rarity. I still think this work is mainly to compare the differences between two different Ginkgo varieties. By the way, you have to provide some cytological evidences for their ploidy.

While for the "High-depth transcriptome ..." in your title, I don't think it is necessary because the sequencing depth of the transcriptome is not related to its genome size. Or you need to compare the differences in the transcriptome with different sequencing depths, but I still think the difference is not significant when the sequencing depth exceeds a certain amount, such as 6G.

I think you can get common genes or gene expression patterns for you have 3 different diploid and haploid ginkgo varieties respectively. And it is more accurate if you conduct the comparing of the common genes or gene expressions between the diploids and the haploids for finding out the differences caused by ploidy.

These are reasons why I asked you to reconstruct your manuscript. By the way, you have to make a proofreading service, such as the charlesworth (https://www.cwauthors.com.cn/).

Round 3

Reviewer 1 Report

In this manuscript, Hu et al. performed morphologic, physiologic and transcriptomic analyses between the haploid and diploid ginkgo. I am thinking that these tiles should be better, "Comparative analysis of morphology, physilogy, and transcriptome between diploid and haploid ginkgo derived from nature" or "Transcriptome reveals differences between diploid and haploid ginkgo derived from nature". It is appropriate to submit it to the IJMS jouranl, but there are still some errors, especially the language. I have pointed out a few (see below), please proofread carefully, including other similar errors that I did not point out. 

line 36-40, please rephrase them. low growth? Why shortening breeding?

line 50-52, inducing "different ploidy" should be "haploids". and please check this paper"Šmarda, P., Horová, L., Knápek, O. et al. Multiple haploids, triploids, and tetraploids found in modern-day “living fossil” Ginkgo biloba. Hortic Res 5, 55 (2018)."

line 52-53, you don't need this here.

line 67, for "the development of ginkgo resources" should be "understanding the effects of gene dosage on gene expressions"

line 70-72, "genomic" should be "transcriptomic" and please be careful for the "first".

line 84, "the material" should be "the haploid and diploid ginkgo cultivars"

line 122-126, please rephrase it more professionally.

line 154-156, be careful  of your conclusion, I suggest you use "might".

line 169-171, be careful of your conclusion

line 176-185, please rephrase them

line 208-209, be careful of your knowledge.

line 219-220, two "still"

line 230, the most "three"

line 233-235, please rephrase them

line 252-253, is this kind of selection for validation scientific? 

line 262-263, please rephrase them

line 265-273, you have to describe more about the haploids, not polyploids for the diploid ginkgo is most often.

line 269, ginkgos or ginkgoes?

line 279, exhibit? apparent or significant?

line 282-284, please rephrase them

line 297-298, please rephrase them

line 299-310, they are nonesense here

line 311-314, be careful  of your conclusion, rephrase them more scientific and professional.

line 328, was?

line 337-342, please rephrase them

line 345-347, please rephrase them

line 348-350, please rephrase them

line 388-391, please rephrase them
